# Learning Kernels with Random Features

**Aman Sinha**[1]    **John Duchi**[1,2]

Departments of [1]Electrical Engineering and [2]Statistics

Stanford University

{amans,jduchi}@stanford.edu

## Abstract

Randomized features provide a computationally efficient way to approximate kernel machines in machine learning tasks. However, such methods require a user-defined kernel as input. We extend the randomized-feature approach to the task of learning a kernel (via its associated random features). Specifically, we present an efficient optimization problem that learns a kernel in a supervised manner. We prove the consistency of the estimated kernel as well as generalization bounds for the class of estimators induced by the optimized kernel, and we experimentally evaluate our technique on several datasets. Our approach is efficient and highly scalable, and we attain competitive results with a fraction of the training cost of other techniques.

## 1 Introduction

An essential element of supervised learning systems is the representation of input data. Kernel methods [27] provide one approach to this problem: they implicitly transform the data to a new feature space, allowing non-linear data representations. This representation comes with a cost, as kernelized learning algorithms require time that grows at least quadratically in the data set size, and predictions with a kernelized procedure require the entire training set. This motivated Rahimi and Recht [24, 25] to develop randomized methods that efficiently approximate kernel evaluations with *explicit* feature transformations; this approach gives substantial computational benefits for large training sets and allows the use of simple linear models in the randomly constructed feature space.

Whether we use standard kernel methods or randomized approaches, using the "right" kernel for a problem can make the difference between learning a useful or useless model. Standard kernel methods as well as the aforementioned randomized-feature techniques assume the input of a user-defined kernel—a weakness if we do not *a priori* know a good data representation. To address this weakness, one often wishes to learn a good kernel, which requires substantial computation. We combine kernel learning with randomization, exploiting the computational advantages offered by randomized features to learn the kernel in a supervised manner. Specifically, we use a simple pre-processing stage for selecting our random features rather than jointly optimizing over the kernel and model parameters. Our workflow is straightforward: we create randomized features, solve a simple optimization problem to select a subset, then train a model with the optimized features. The procedure results in lower-dimensional models than the original random-feature approach for the same performance. We give empirical evidence supporting these claims and provide theoretical guarantees that our procedure is consistent with respect to the limits of infinite training data and infinite-dimensional random features.

### 1.1 Related work

To discuss related work, we first describe the supervised learning problem underlying our approach. We have a cost $c : \mathbb{R} \times \mathcal{Y} \to \mathbb{R}$, where $c(\cdot, y)$ is convex for $y \in \mathcal{Y}$, and a reproducing kernel Hilbert space (RKHS) of functions $\mathcal{F}$ with kernel $K$. Given a sample $\{(x^i, y^i)\}_{i=1}^n$, the usual $\ell_2$-regularized

learning problem is to solve the following (shown in primal and dual forms respectively):

$$\underset{f \in \mathcal{F}}{\text{minimize}} \quad \sum_{i=1}^{n} c(f(x^i), y^i) + \frac{\lambda}{2} \|f\|_2^2, \quad \text{or} \quad \underset{\alpha \in \mathbb{R}^n}{\text{maximize}} \quad -\sum_{i=1}^{n} c^*(\alpha_i, y^i) - \frac{1}{2\lambda} \alpha^T G \alpha, \quad (1)$$

where $\|\cdot\|_2$ denotes the Hilbert space norm, $c^*(\alpha, y) = \sup_z \{\alpha z - c(z, y)\}$ is the convex conjugate of $c$ (for fixed $y$) and $G = [K(x^i, x^j)]_{i,j=1}^n$ denotes the Gram matrix.

Several researchers have studied kernel learning. As noted by Gönen and Alpaydın [14], most formulations fall into one of a few categories. In the supervised setting, one assumes a base class or classes of kernels and either uses heuristic rules to combine kernels [2, 23], optimizes structured (e.g. linear, nonnegative, convex) compositions of the kernels with respect to an alignment metric [9, 16, 20, 28], or jointly optimizes kernel compositions with empirical risk [17, 20, 29]. The latter approaches require an eigendecomposition of the Gram matrix or costly optimization problems (e.g. quadratic or semidefinite programs) [10, 14], but these models have a variety of generalization guarantees [1, 8, 10, 18, 19]. Bayesian variants of compositional kernel search also exist [12, 13]. In un- and semi-supervised settings, the goal is to learn an embedding of the input distribution followed by a simple classifier in the embedded space (e.g. [15]); the hope is that the input distribution carries the structure relevant to the task. Despite the current popularity of these techniques, especially deep neural architectures, they are costly, and it is difficult to provide guarantees on their performance.

Our approach optimizes kernel compositions with respect to an alignment metric, but rather than work with Gram matrices in the original data representation, we work with randomized feature maps that approximate RKHS embeddings. We learn a kernel that is structurally different from a user-supplied base kernel, and our method is an efficiently (near linear-time) solvable convex program.

## 2 Proposed approach

At a high level, we take a feature mapping, find a distribution that aligns this mapping with the labels $y$, and draw random features from the learned distribution; we then use these features in a standard supervised learning approach.

For simplicity, we focus on binary classification: we have $n$ datapoints $(x^i, y^i) \in \mathbb{R}^d \times \{-1, 1\}$. Letting $\phi : \mathbb{R}^d \times \mathcal{W} \to [-1, 1]$ and $Q$ be a probability measure on a space $\mathcal{W}$, define the kernel

$$K_Q(x, x') := \int \phi(x, w)\phi(x', w) dQ(w). \quad (2)$$

We want to find the "best" kernel $K_Q$ over all distributions $Q$ in some (large, nonparametric) set $\mathcal{P}$ of possible distributions on random features; we consider a kernel alignment problem of the form

$$\underset{Q \in \mathcal{P}}{\text{maximize}} \quad \sum_{i,j} K_Q(x^i, x^j) y^i y^j. \quad (3)$$

We focus on sets $\mathcal{P}$ defined by divergence measures on the space of probability distributions. For a convex function $f$ with $f(1) = 0$, the $f$-divergence between distributions $P$ and $Q$ is $D_f(P\|Q) = \int f(\frac{dP}{dQ}) dQ$. Then, for a base (user-defined) distribution $P_0$, we consider collections $\mathcal{P} := \{Q : D_f(Q\|P_0) \le \rho\}$ where $\rho > 0$ is a specified constant. In this paper, we focus on divergences $f(t) = t^k - 1$ for $k \ge 2$. Intuitively, the distribution $Q$ maximizing the alignment (3) gives a feature space in which pairwise distances are similar to those in the output space $\mathcal{Y}$. Unfortunately, the problem (3) is generally intractable as it is infinite dimensional.

Using the randomized feature approach, we approximate the integral (2) as a discrete sum over samples $W^i \overset{\text{iid}}{\sim} P_0$, $i \in [N_w]$. Defining the discrete approximation $\mathcal{P}_{N_w} := \{q : D_f(q\|\mathbf{1}/N_w) \le \rho\}$ to $\mathcal{P}$, we have the following empirical version of problem (3):

$$\underset{q \in \mathcal{P}_{N_w}}{\text{maximize}} \quad \sum_{i,j} y^i y^j \sum_{m=1}^{N_w} q_m \phi(x^i, w^m) \phi(x^j, w^m). \quad (4)$$

Using randomized features, matching the input and output distances in problem (4) translates to finding a (weighted) set of points among $w^1, w^2, ..., w^{N_w}$ that best "describe" the underlying dataset, or, more directly, finding weights $q$ so that the kernel matrix matches the correlation matrix $yy^T$.

Given a solution $\widehat{q}$ to problem (4), we can solve the primal form of problem (1) in two ways. First, we can apply the Rahimi and Recht [24] approach by drawing $D$ samples $W^1, \ldots, W^D \overset{\text{iid}}{\sim} \widehat{q}$, defining features $\phi^i = [\phi(x^i, w^1) \ \cdots \ \phi(x^i, w^D)]^T$, and solving the risk minimization problem

$$\widehat{\theta} = \underset{\theta}{\operatorname{argmin}} \left\{ \sum_{i=1}^n c\left(\tfrac{1}{\sqrt{D}}\theta^T \phi^i, y^i\right) + r(\theta) \right\}$$

(5)

for some regularization $r$. Alternatively, we may set $\phi^i = [\phi(x^i, w^1) \ \cdots \ \phi(x^i, w^{N_w})]^T$, where $w^1, \ldots, w^{N_w}$ are the original random samples from $P_0$ used to solve (4), and directly solve

$$\widehat{\theta} = \underset{\theta}{\operatorname{argmin}} \left\{ \sum_{i=1}^n c(\theta^T \operatorname{diag}(\widehat{q})^{\frac{1}{2}} \phi^i, y^i) + r(\theta) \right\}.$$

(6)

Notably, if $\widehat{q}$ is sparse, the problem (6) need only store the random features corresponding to non-zero entries of $\widehat{q}$. Contrast our two-phase procedure to that of Rahimi and Recht [25], which samples $W^1, \ldots, W^D \overset{\text{iid}}{\sim} P_0$ and solves the minimization problem

$$\underset{\alpha \in \mathbb{R}^{N_w}}{\operatorname{minimize}} \ \sum_{i=1}^n c\left( \sum_{m=1}^D \alpha_m \phi(x^i, w^m), y^i \right) \ \text{ subject to } \ \|\alpha\|_\infty \leq C/N_w,$$

(7)

where $C$ is a numerical constant. At first glance, it appears that we may suffer both in terms of computational efficiency and in classification or learning performance compared to the one-step procedure (7). However, as we show in the sequel, the alignment problem (4) can be solved very efficiently and often yields sparse vectors $\widehat{q}$, thus substantially decreasing the dimensionality of problem (6). Additionally, we give experimental evidence in Section 4 that the two-phase procedure yields generalization performance similar to standard kernel and randomized feature methods.

## 2.1 Efficiently solving problem (4)

The optimization problem (4) has structure that enables efficient (near linear-time) solutions. Define the matrix $\Phi = [\phi^1 \ \cdots \ \phi^n] \in \mathbb{R}^{N_w \times n}$, where $\phi^i = [\phi(x^i, w^1) \ \cdots \ \phi(x^i, w^{N_w})]^T \in \mathbb{R}^{N_w}$ is the randomized feature representation for $x^i$ and $w^m \overset{\text{iid}}{\sim} P_0$. We can rewrite the optimization objective as

$$\sum_{i,j} y^i y^j \sum_{m=1}^{N_w} q_m \phi(x^i, w^m)\phi(x^j, w^m) = \sum_{m=1}^{N_w} q_m \left( \sum_{i=1}^n y^i \phi(x^i, w^m) \right)^2 = q^T \left( (\Phi y) \odot (\Phi y) \right),$$

where $\odot$ denotes the Hadamard product. Constructing the linear objective requires the evaluation of $\Phi y$. Assuming that the computation of $\phi$ is $O(d)$, construction of $\Phi$ is $O(n N_w d)$ on a single processor. However, this construction is trivially parallelizable. Furthermore, computation can be sped up even further for certain distributions $P_0$. For example, the Fastfood technique can approximate $\Phi$ in $O(n N_w \log(d))$ time for the Gaussian kernel [21].

The problem (4) is also efficiently solvable via bisection over a scalar dual variable. Using $\lambda \geq 0$ for the constraint $D_f\left(Q\|P_0\right) \leq \rho$, a partial Lagrangian is

$$\mathcal{L}(q, \lambda) = q^T \left( (\Phi y) \odot (\Phi y) \right) - \lambda \left( D_f\left(q\|\mathbf{1}/N_w\right) - \rho \right).$$

The corresponding dual function is $g(\lambda) = \sup_{q \in \Delta} \mathcal{L}(q, \lambda)$, where $\Delta := \{q \in \mathbb{R}_+^{N_w} : q^T \mathbf{1} = 1\}$ is the probability simplex. Minimizing $g(\lambda)$ yields the solution to problem (4); this is a convex optimization problem in one dimension so we can use bisection. The computationally expensive step in each iteration is maximizing $\mathcal{L}(q, \lambda)$ with respect to $q$ for a given $\lambda$. For $f(t) = t^k - 1$, we define $v := (\Phi y) \odot (\Phi y)$ and solve

$$\underset{q \in \Delta}{\operatorname{maximize}} \ q^T v - \lambda \frac{1}{N_w} \sum_{m=1}^{N_w} (N_w q_m)^k.$$

(8)

This has a solution of the form $q_m = \left[ v_m/\lambda N_w^{k-1} + \tau \right]_+^{\frac{1}{k-1}}$, where $\tau$ is chosen so that $\sum_m q_m = 1$. We can find such a $\tau$ by a variant of median-based search in $O(N_w)$ time [11]. Thus, for any $k \geq 2$, an $\epsilon$-suboptimal solution to problem (4) can be found in $O(N_w \log(1/\epsilon))$ time (see Algorithm 1).

**Algorithm 1** Kernel optimization with $f(t) = t^k - 1$ as divergence

---

INPUT: distribution $P_0$ on $\mathcal{W}$, sample $\{(x^i, y^i)\}_{i=1}^n$, $N_w \in \mathbb{N}$, feature function $\phi$, $\epsilon > 0$
OUTPUT: $q \in \mathbb{R}^{N_w}$ that is an $\epsilon$-suboptimal solution to (4).
SETUP: Draw $N_w$ samples $w^m \overset{\text{iid}}{\sim} P_0$, build feature matrix $\Phi$, compute $v := (\Phi y) \odot (\Phi y)$.
Set $\lambda_u \leftarrow \infty$, $\lambda_l \leftarrow 0$, $\lambda_s \leftarrow 1$
**while** $\lambda_u = \infty$
   $q \leftarrow \text{argmax}_{q \in \Delta} \mathcal{L}(q, \lambda_s)$     *// (solution to problem (8))*
   **if** $D_f(q \| 1/N_w) < \rho$ **then**    $\lambda_u \leftarrow \lambda_s$    **else** $\lambda_s \leftarrow 2\lambda_s$
**while** $\lambda_u - \lambda_l > \epsilon \lambda_s$
   $\lambda \leftarrow (\lambda_u + \lambda_l)/2$
   $q \leftarrow \text{argmax}_{q \in \Delta} \mathcal{L}(q, \lambda)$     *// (solution to problem (8))*
   **if** $D_f(q \| 1/N_w) < \rho$ **then**    $\lambda_u \leftarrow \lambda$    **else** $\lambda_l \leftarrow \lambda$

---

## 3 Consistency and generalization performance guarantees

Although the procedure (4) is a discrete approximation to a heuristic kernel alignment problem, we can provide guarantees on its performance as well as the generalization performance of our subsequent model trained with the optimized kernel.

**Consistency** First, we provide guarantees that the solution to problem (4) approaches a population optimum as the data and random sampling increase ($n \to \infty$ and $N_w \to \infty$, respectively). We consider the following (slightly more general) setting: let $S : \mathcal{X} \times \mathcal{X} \to [-1, 1]$ be a bounded function, where we intuitively think of $S(x, x')$ as a similarity metric between labels for $x$ and $x'$, and denote $S_{ij} := S(x^i, x^j)$ (in the binary case with $y \in \{-1, 1\}$, we have $S_{ij} = y^i y^j$). We then define the alignment functions

$$T(P) := \mathbb{E}[S(X, X')K_P(X, X')], \quad \widehat{T}(P) := \frac{1}{n(n-1)} \sum_{i \neq j} S_{ij} K_P(x^i, x^j),$$

where the expectation is taken over $S$ and the independent variables $X, X'$. Lemmas 1 and 2 provide consistency guarantees with respect to the data sample ($x^i$ and $S_{ij}$) and the random feature sample ($w^m$); together they give us the overall consistency result of Theorem 1. We provide proofs in the supplement (Sections A.1, A.2, and A.3 respectively).

**Lemma 1** (Consistency with respect to data). *Let $f(t) = t^k - 1$ for $k \geq 2$. Let $P_0$ be any distribution on the space $\mathcal{W}$, and let $\mathcal{P} = \{Q : D_f(Q \| P_0) \leq \rho\}$. Then*

$$\mathbb{P}\left( \sup_{Q \in \mathcal{P}} \left| \widehat{T}(Q) - T(Q) \right| \geq t \right) \leq \sqrt{2} \exp\left( -\frac{nt^2}{16(1+\rho)} \right).$$

Lemma 1 shows that the empirical quantity $\widehat{T}$ is close to the true $T$. Now we show that, independent of the size of the training data, we can consistently estimate the optimal $Q \in \mathcal{P}$ via sampling (i.e. $Q \in \mathcal{P}_{N_w}$).

**Lemma 2** (Consistency with respect to sampling features). *Let the conditions of Lemma 1 hold. Then, with $C_\rho = \frac{2(\rho+1)}{\sqrt{1+\rho}-1}$ and $D_\rho = \sqrt{8(1+\rho)}$, we have*

$$\left| \sup_{Q \in \mathcal{P}_{N_w}} \widehat{T}(Q) - \sup_{Q \in \mathcal{P}} \widehat{T}(Q) \right| \leq 4 C_\rho \sqrt{\frac{\log(2N_w)}{N_w}} + D_\rho \sqrt{\frac{\log \frac{2}{\delta}}{N_w}}$$

*with probability at least $1 - \delta$ over the draw of the samples $W^m \overset{\text{iid}}{\sim} P_0$.*

Finally, we combine the consistency guarantees for data and sampling to reach our main result, which shows that the alignment provided by the estimated distribution $\widehat{Q}$ is nearly optimal.

**Theorem 1.** *Let $\widehat{Q}_w$ maximize $\widehat{T}(Q)$ over $Q \in \mathcal{P}_{N_w}$. Then, with probability at least $1 - 3\delta$ over the sampling of both $(x, y)$ and $W$, we have*

$$\left| T(\widehat{Q}_w) - \sup_{Q \in \mathcal{P}} T(Q) \right| \leq 4 C_\rho \sqrt{\frac{\log(2N_w)}{N_w}} + D_\rho \sqrt{\frac{\log \frac{2}{\delta}}{N_w}} + 2 D_\rho \sqrt{\frac{2 \log \frac{2}{\delta}}{n}}.$$

**Generalization performance**   The consistency results above show that our optimization procedure nearly maximizes alignment $T(P)$, but they say little about generalization performance for our model trained using the optimized kernel. We now show that the class of estimators employed by our method has strong performance guarantees. By construction, our estimator (6) uses the function class

$$\mathcal{F}_{N_w} := \Big\{ h(x) = \sum_{m=1}^{N_w} \alpha_m \sqrt{q_m} \phi(x, w^m) \mid q \in \mathcal{P}_{N_w}, \|\alpha\|_2 \leq B \Big\},$$

and we provide bounds on its generalization via empirical Rademacher complexity. To that end, define $\mathcal{R}_n(\mathcal{F}_{N_w}) := \frac{1}{n}\mathbb{E}[\sup_{f \in \mathcal{F}_{N_w}} \sum_{i=1}^n \sigma_i f(x^i)]$, where the expectation is taken over the i.i.d. Rademacher variables $\sigma_i \in \{-1, 1\}$. We have the following lemma, whose proof is in Section A.4.

**Lemma 3.** *Under the conditions of the preceding paragraph, $\mathcal{R}_n(\mathcal{F}_{N_w}) \leq B\sqrt{\frac{2(1+\rho)}{n}}$.*

Applying standard concentration results, we obtain the following generalization guarantee.

**Theorem 2** ([8, 18]). *Let the true misclassification risk and $\nu$-empirical misclassification risk for an estimator $h$ be defined as follows:*

$$R(h) := \mathbb{P}(Yh(X) < 0), \quad \widehat{R}_\nu(h) := \frac{1}{n}\sum_{i=1}^n \min\Big\{1, \big[1 - yh(x^i)/\nu\big]_+\Big\}.$$

*Then $\sup_{h \in \mathcal{F}_{N_w}} \{R(h) - \widehat{R}_\nu(h)\} \leq \frac{2}{\nu}\mathcal{R}_n(\mathcal{F}_{N_w}) + 3\sqrt{\frac{\log\frac{2}{\delta}}{2n}}$ with probability at least $1 - \delta$.*

The bound is independent of the number of terms $N_w$, though in practice we let $B$ grow with $N_w$.

# 4   Empirical evaluations

We now turn to empirical evaluations, comparing our approach's predictive performance with that of Rahimi and Recht's randomized features [24] as well as a joint optimization over kernel compositions and empirical risk. In each of our experiments, we investigate the effect of increasing dimensionality of the randomized feature space $D$. For our approach, we use the $\chi^2$-divergence ($k = 2$ or $f(t) = t^2 - 1$). Letting $\widehat{q}$ denote the solution to problem (4), we use two variants of our approach: when $D < \mathrm{nnz}(\widehat{q})$ we use estimator (5), and we use estimator (6) otherwise. For the original randomized feature approach, we relax the constraint in problem (7) with an $\ell_2$ penalty. Finally, for the joint optimization in which we learn the kernel and classifier together, we consider the kernel-learning objective, i.e. finding the best Gram matrix $G$ in problem (1) for the soft-margin SVM [14]:

$$\begin{aligned}\text{minimize}_{q \in \mathcal{P}_{N_w}} \quad &\sup_\alpha \quad \alpha^T\mathbf{1} - \frac{1}{2}\sum_{i,j}\alpha_i\alpha_j y^i y^j \sum_{m=1}^{N_w} q_m \phi(x^i, w^m)\phi(x^j, w^m) \\ &\text{subject to} \quad \mathbf{0} \preceq \alpha \preceq C\mathbf{1}, \quad \alpha^T y = 0.\end{aligned} \quad (9)$$

We use a standard primal-dual algorithm [4] to solve the min-max problem (9). While this is an expensive optimization, it is a convex problem and is solvable in polynomial time.

In Section 4.1, we visualize a particular problem that illustrates the effectiveness of our approach when the user-defined kernel is poor. Section 4.2 shows how learning the kernel can be used to quickly find a sparse set of features in high dimensional data, and Section 4.3 compares our performance with unoptimized random features and the joint procedure (9) on benchmark datasets. The supplement contains more experimental results in Section C.

## 4.1   Learning a new kernel with a poor choice of $P_0$

For our first experiment, we generate synthetic data $x^i \stackrel{\text{iid}}{\sim} \mathsf{N}(0, I)$ with labels $y^i = \mathrm{sign}(\|x\|_2 - \sqrt{d})$, where $x \in \mathbb{R}^d$. The Gaussian kernel is ill-suited for this task, as the Euclidean distance used in this kernel does not capture the underlying structure of the classes. Nevertheless, we use the Gaussian kernel, which corresponds [24] to $\phi(x, (w, v)) = \cos((x, 1)^T(w, v))$ where $(W, V) \sim \mathsf{N}(0, I) \times \mathsf{Uni}(0, 2\pi)$, to showcase the effects of our method. We consider a training set of size $n = 10^4$ and a test set of size $10^3$, and we employ logistic regression with $D = \mathrm{nnz}(\widehat{q})$ for both our technique as well as the original random feature approach.[1]

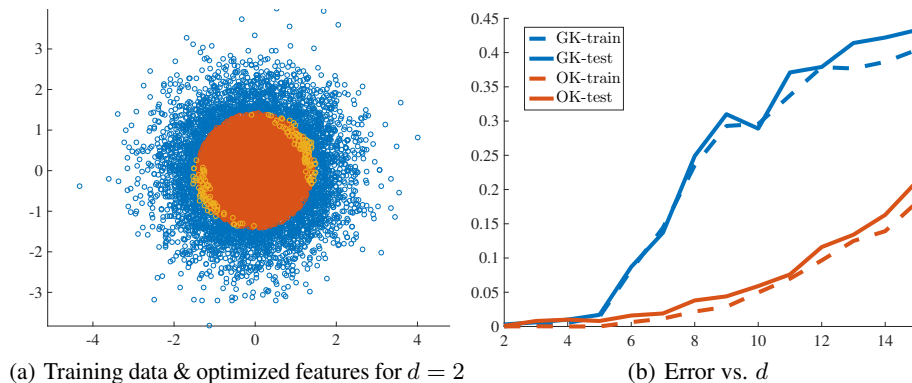

(a) Training data & optimized features for $d = 2$    (b) Error vs. $d$

**Figure 1.** Experiments with synthetic data. (a) Positive and negative training examples are blue and red, and optimized randomized features ($w^m$) are yellow. All offset parameters $v^m$ were optimized to be near 0 or $\pi$ (not shown). (b) Misclassification error of logistic regression model vs. dimensionality of data. GK denotes random features with a Gaussian kernel, and our optimized kernel is denoted OK.

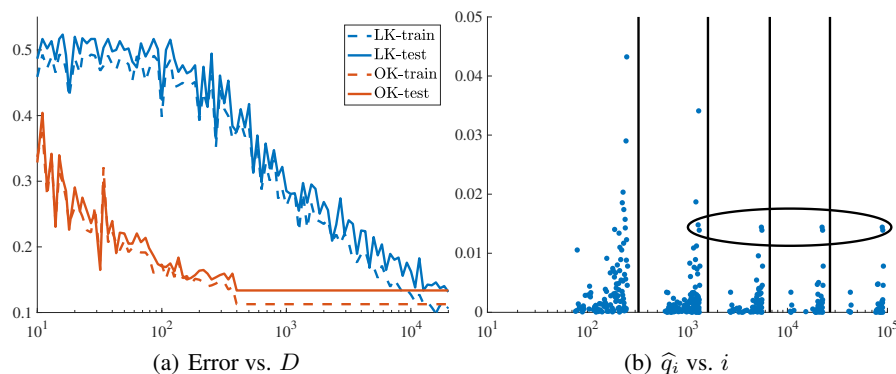

(a) Error vs. $D$    (b) $\widehat{q}_i$ vs. $i$

**Figure 2.** Feature selection in sparse data. (a) Misclassification error of ridge regression model vs. dimensionality of data. LK denotes random features with a linear kernel, and OK denotes our method. Our error is fixed above $D = \text{nnz}(\widehat{q})$ after which we employ estimator (6). (b) Weight of feature $i$ in optimized kernel ($q_i$) vs. $i$. Vertical bars delineate separations between $k$-grams, where $1 \leq k \leq 5$ is nondecreasing in $i$. Circled features are prefixes of `GGTTG` and `GTTGG` at indices 60–64.

Figure 1 shows the results of the experiments for $d \in \{2, \dots, 15\}$. Figure 1(a) illustrates the output of the optimization when $d = 2$. The selected kernel features $w^m$ lie near $(1, 1)$ and $(-1, -1)$; the offsets $v^m$ are near 0 and $\pi$, giving the feature $\phi(\cdot, w, v)$ a parity flip. Thus, the kernel computes similarity between datapoints via neighborhoods of $(1, 1)$ and $(-1, -1)$ close to the classification boundary. In higher dimensions, this generalizes to neighborhoods of pairs of opposing points along the surface of the $d$-sphere; these features provide a coarse approximation to vector magnitude. Performance degradation with $d$ occurs because the neighborhoods grow exponentially larger and less dense (due to fixed $N_w$ and $n$). Nevertheless, as shown in Figure 1(b), this degradation occurs much more slowly than that of the Gaussian kernel, which suffers a similar curse of dimensionality due to its dependence on Euclidean distance. Although somewhat contrived, this example shows that even in situations with poor base kernels our approach learns a more suitable representation.

## 4.2 Feature selection and biological sequences

In addition to the computational advantages rendered by the sparsity of $q$ after performing the optimization (4), we can use this sparsity to gain insights about important features in high-dimensional datasets; this can act as an efficient filtering mechanism before further investigation. We present one example of this task, studying an aptamer selection problem [6]. In this task, we are given $n = 2900$ nucleotide sequences (aptamers) $x^i \in \mathcal{A}^{81}$, where $\mathcal{A} = \{\text{A}, \text{C}, \text{G}, \text{T}\}$ and labels $y^i$ indicate (thresholded) binding affinity of the aptamer to a molecular target. We create one-hot encoded forms of $k$-grams of the sequence, where $1 \leq k \leq 5$, resulting in $d = \sum_{k=1}^{5} |\mathcal{A}|^k (82 - k) = 105{,}476$

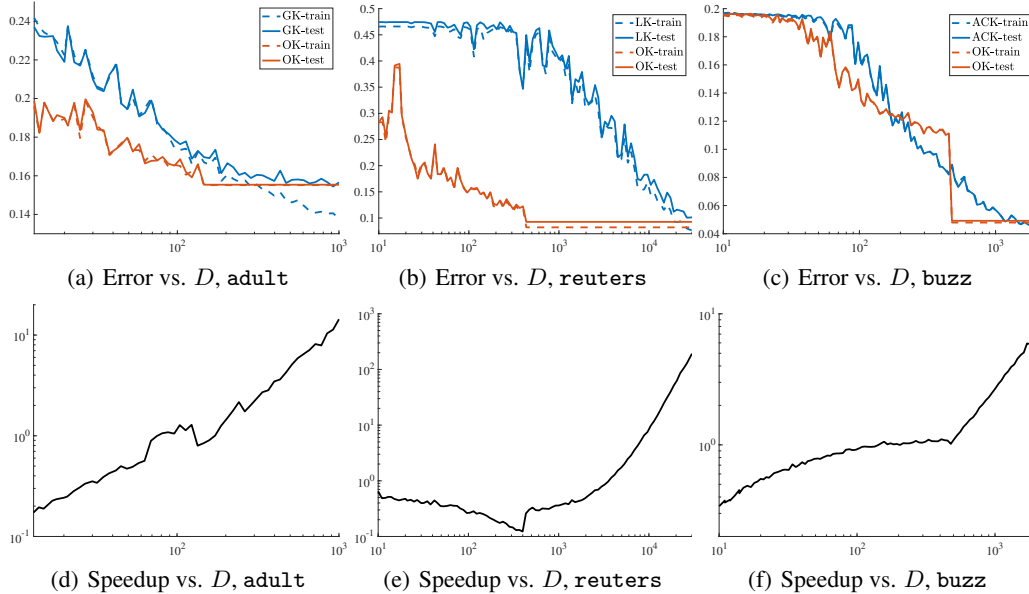

(a) Error vs. $D$, `adult`  (b) Error vs. $D$, `reuters`  (c) Error vs. $D$, `buzz`

(d) Speedup vs. $D$, `adult`  (e) Speedup vs. $D$, `reuters`  (f) Speedup vs. $D$, `buzz`

**Figure 3.** Performance analysis on benchmark datasets. The top row shows training and test misclassification rates. Our method is denoted as OK and is shown in red. The blue methods are random features with Gaussian, linear, or arc-cosine kernels (GK, LK, or ACK respectively). Our error and running time become fixed above $D = \text{nnz}(\widehat{q})$ after which we employ estimator (6). The bottom row shows the speedup factor of using our method over regular random features (speedup $= x$ indicates our method takes $1/x$ of the time required to use regular random features). Our method is faster at moderate to large $D$ and shows better performance than the random feature approach at small to moderate $D$.

**Table 1:** Best test results over benchmark datasets

| Dataset | $n,$ | $n_{test}$ | $d$ | Model | Our error (%), time(s) | | Random error (%), time(s) | |
|---|---|---|---|---|---|---|---|---|
| `adult` | 32561, | 16281 | 123 | Logistic | 15.54, | 3.6 | 15.44, | 43.1 |
| `reuters` | 23149, | 781265 | 47236 | Ridge | 9.27, | 0.8 | 9.36, | 295.9 |
| `buzz` | 105530, | 35177 | 77 | Ridge | 4.92, | 2.0 | 4.58, | 11.9 |

features. We consider the linear kernel, i.e. $\phi(x, w) = x_w$, where $w \sim \text{Uni}(\{1, \dots, d\})$. Figure 2(a) compares the misclassification error of our method with that of random $k$-gram features, while Figure 2(b) indicates the weights $q_i$ given to features by our method. In under $0.2$ seconds, we whittle down the original feature space to 379 important features. By restricting random selection to just these features, we outperform the approach of selecting features uniformly at random when $D \ll d$. More importantly, however, we can derive insights from this selection. For example, the circled features in Figure 2(b) correspond to $k$-gram prefixes for the 5-grams `GGTTG` and `GTTGG` at indices 60 through 64; `G`-complexes are known to be relevant for binding affinities in aptamers [6], so this is reasonable.

## 4.3 Performance on benchmark datasets

We now show the benefits of our approach on large-scale datasets, since we exploit the efficiency of random features with the performance of kernel-learning techniques. We perform experiments on three distinct types of datasets, tracking training/test error rates as well as total (training + test) time. For the `adult`[2] dataset we employ the Gaussian kernel with a logistic regression model, and for the `reuters`[3] dataset we employ a linear kernel with a ridge regression model. For the `buzz`[4] dataset we employ ridge regression with an arc-cosine kernel of order 2, i.e. $P_0 = \mathcal{N}(0, I)$ and $\phi(x, w) = H(w^T x)(w^T x)^2$, where $H(\cdot)$ is the Heaviside step function [7].

**Table 2:** Comparisons with joint optimization on subsampled data

| Dataset | Our training / test error (%), time($s$) | | | Joint training / test error (%), time($s$) | | |
|---|---|---|---|---|---|---|
| adult | 16.22 | / | 16.36, | 1.8 | 14.88 / 16.31, | 198.1 |
| reuters | 7.64 | / | 9.66, | 0.6 | 6.30 / 8.96, | 173.3 |
| buzz | 8.44 | / | 8.32, | 0.4 | 7.38 / 7.08, | 137.5 |

**Comparison with unoptimized random features**   Results comparing our method with unoptimized random features are shown in Figure 3 for many values of $D$, and Table 1 tabulates the best test error and corresponding time for the methods. Our method outperforms the original random feature approach in terms of generalization error for small and moderate values of $D$; at very large $D$ the random feature approach either matches our surpasses our performance. The trends in speedup are opposite: our method requires extra optimizations that dominate training time at extremely small $D$; at very large $D$ we use estimator (6), so our method requires less overall time. The nonmonotonic behavior for reuters (Figure 3(e)) occurs due to the following: at $D \lesssim \mathrm{nnz}(\widehat{q})$, sampling indices from the optimized distribution takes a non-neglible fraction of total time, and solving the linear system requires more time when rows of $\Phi$ are not unique (due to sampling).

Performance improvements also depend on the kernel choice for a dataset. Namely, our method provides the most improvement, in terms of training time for a given amount of generalization error, over random features generated for the linear kernel on the reuters dataset; we are able to surpass the best results of the random feature approach 2 orders of magnitude faster. This makes sense when considering the ability of our method to sample from a small subset of important features. On the other hand, random features for the arc-cosine kernel are able to achieve excellent results on the buzz dataset even without optimization, so our approach only offers modest improvement at small to moderate $D$. For the Gaussian kernel employed on the adult dataset, our method is able to achieve the same generalization performance as random features in roughly $1/12$ the training time.

Thus, we see that our optimization approach generally achieves competitive results with random features at lower computational costs, and it offers the most improvements when either the base kernel is not well-suited to the data or requires a large number of random features (large $D$) for good performance. In other words, our method reduces the sensitivity of model performance to the user's selection of base kernels.

**Comparison with joint optimization**   Despite the fact that we do not choose empirical risk as our objective in optimizing kernel compositions, our optimized kernel enjoys competitive generalization performance compared to the joint optimization procedure (9). Because the joint optimization is very costly, we consider subsampled training datasets of 5000 training examples. Results are shown in Table 2, where it is evident that the efficiency of our method outweighs the marginal gain in classification performance for joint optimization.

## 5   Conclusion

We have developed a method to learn a kernel in a supervised manner using random features. Although we consider a kernel alignment problem similar to other approaches in the literature, we exploit computational advantages offered by random features to develop a much more efficient and scalable optimization procedure. Our concentration bounds guarantee the results of our optimization procedure closely match the limits of infinite data ($n \rightarrow \infty$) and sampling ($N_w \rightarrow \infty$), and our method produces models that enjoy good generalization performance guarantees. Empirical evaluations indicate that our optimized kernels indeed "learn" structure from data, and we attain competitive results on benchmark datasets at a fraction of the training time for other methods. Generalizing the theoretical results for concentration and risk to other $f-$divergences is the subject of further research. More broadly, our approach opens exciting questions regarding the usefulness of simple optimizations on random features in speeding up other traditionally expensive learning problems.

**Acknowledgements**   This research was supported by a Fannie & John Hertz Foundation Fellowship and a Stanford Graduate Fellowship.

## Footnotes

[1]For $2 \leq d \leq 15$, $\mathrm{nnz}(\widehat{q}) < 250$ when the kernel is trained with $N_w = 2 \cdot 10^4$, and $\rho = 200$.

[2]https://archive.ics.uci.edu/ml/datasets/Adult

[3]http://www.ai.mit.edu/projects/jmlr/papers/volume5/lewis04a/lyrl2004_rcv1v2_README.htm. We consider predicting whether a document has a CCAT label.

[4]http://ama.liglab.fr/data/buzz/classification/. We use the Twitter dataset.

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
