[Supplementary Material · nips2016supplement.pdf]

# A  Proofs of major results

Before proving our results, we provide a few technical lemmas to which we refer in the sequel, and we also give a few definitions. The first is the standard definition of sub-Gaussian random variables.

**Definition 1.** *A random variable $X$ is $\sigma^2$-sub-Gaussian if*

$$\mathbb{E}\left[\exp(\lambda(X - \mathbb{E}[X]))\right] \leq \exp\left(\frac{\lambda^2\sigma^2}{2}\right)$$

*for all $\lambda \in \mathbb{R}$.*

We enumerate a few standard consequences of sub-Gaussianity [5]. If $X_i$ are independent and $\sigma^2$-sub-Gaussian, then $\sum_{i=1}^n X_i$ is $n\sigma^2$-sub-Gaussian. Moreover, we have the standard concentration guarantee

$$\max\{\mathbb{P}(X \geq \mathbb{E}[X] + t), \mathbb{P}(X \leq \mathbb{E}[X] - t)\} \leq 2\exp\left(-\frac{t^2}{2\sigma^2}\right)$$

for all $t \geq 0$ if $X$ is $\sigma^2$-sub-Gaussian, and if there are bounds $a \leq X \leq b$, then $X$ is $\frac{(b-a)^2}{4}$-sub-Gaussian. Moreover, if $X$ is mean-zero and $\sigma^2$-sub-Gaussian, then

$$\mathbb{E}\left[\exp(\lambda X^2)\right] \leq \frac{1}{[1 - 2\lambda\sigma^2]_+^{\frac{1}{2}}} = \exp\left(-\frac{1}{2}\log\left[1 - 2\lambda\sigma^2\right]_+\right). \tag{10}$$

Throughout our proofs, for a given $k \in [1, \infty]$, we use $k_* = \frac{k}{k-1}$, so that $1/k + 1/k_* = 1$, to denote the conjugate to $k$.

The technical lemmas that we shall need follow. The first is an essentially standard duality result.

**Lemma 4** (Ben-Tal et al. [3])**.** *Let $f$ be any closed convex function with domain $\operatorname{dom} f \subset [0, \infty)$, and let $f^*(s) = \sup_{t \geq 0}\{ts - f(t)\}$ be its conjugate. Then for any distribution $P$ and any function $g : \mathcal{W} \to \mathbb{R}$ we have*

$$\sup_{Q:D_f(Q\|P)\leq\rho} \int g(w)dQ(w) = \inf_{\lambda\geq0,\eta}\left\{\lambda\int f^*\left(\frac{g(w)-\eta}{\lambda}\right)dP(w) + \rho\lambda + \eta\right\}.$$

See Section B.1 for a proof of this lemma. Note that as an immediate consequence of this result, we have an expectation upper bound on empirical versions of $\sup_{Q:D_f(Q\|P)\leq\rho} \int g(w)dQ(w)$. Indeed, let $Z_1, \ldots, Z_{N_w}$ be drawn i.i.d. from a base distribution $P_0$. To simplify algebra, we work with a scaled version of the $f$-divergence: $f(t) = \frac{1}{k}(t^k - 1)$, so the population and empirical constraint sets we consider are defined by

$$\mathcal{P} = \left\{Q : D_f\left(Q\|P_0\right) \leq \frac{\rho}{k}\right\} \text{ and } \mathcal{P}_{N_w} := \left\{q : D_f\left(q\|\mathbf{1}/N_w\right) \leq \frac{\rho}{k}\right\}.$$

Then by Lemma 4, we obtain

$$
\begin{aligned}
\mathbb{E}\left[\sup_{Q\in\mathcal{P}_{N_w}} \mathbb{E}_Q[Z]\right] &= \mathbb{E}_{P_0}\left[\inf_{\lambda\geq0,\eta} \frac{1}{N}\sum_{i=1}^N \lambda f^*\left(\frac{Z_i-\eta}{\lambda}\right) + \eta + \frac{\rho}{k}\lambda\right] \\
&\leq \inf_{\lambda\geq0,\eta} \mathbb{E}_{P_0}\left[\frac{1}{N}\sum_{i=1}^N \lambda f^*\left(\frac{Z_i-\eta}{\lambda}\right) + \eta + \frac{\rho}{k}\lambda\right] \\
&= \inf_{\lambda\geq0,\eta}\left\{\mathbb{E}_{P_0}\left[\lambda f^*\left(\frac{Z-\eta}{\lambda}\right)\right] + \frac{\rho}{k}\lambda + \eta\right\} \\
&= \sup_{Q\in\mathcal{P}} \mathbb{E}_Q[Z]. \tag{11}
\end{aligned}
$$

The second lemma provides a lower bound on the expectation of certain robust quantities, and we provide a proof of the lemma in Section B.2.

**Lemma 5.** *Let $Z = (Z_1, \ldots, Z_{N_w})$ be a random vector of independent random variables $Z_i \overset{\text{iid}}{\sim} P_0$, where $|Z_i| \leq M$ with probability $1$. Let $k \in [2, \infty]$ and define $C_{\rho,k} = \frac{2(1+\rho)}{(1+\rho)^{\frac{1}{k_*}} - 1} \leq C_\rho = \frac{2(\rho+1)}{\sqrt{1+\rho}-1}$. Let $f(t) = \frac{1}{k}(t^k - 1)$. Then*

$$\mathbb{E}\left[\sup_{Q \in \mathcal{P}_{N_w}} \mathbb{E}_Q[Z]\right] \geq \sup_{Q \in \mathcal{P}} \mathbb{E}_Q[Z] - 4C_\rho M \sqrt{\frac{\log(2N_w)}{N_w}}$$

*and*

$$\mathbb{E}\left[\sup_{Q \in \mathcal{P}_{N_w}} \mathbb{E}_Q[Z]\right] \leq \sup_{Q \in \mathcal{P}} \mathbb{E}_Q[Z].$$

### A.1 Proof of Lemma 1

The result follows from a dual formulation of the expression on the left hand side as well as standard concentration results for sub-Gaussian random variables. Define

$$\widehat{e}_n(w) := \frac{1}{n(n-1)} \sum_{i \neq j} S_{ij} \phi(x^i, w) \phi(x^j, w) - \mathbb{E}[S(X, X')\phi(X, w)\phi(X', w)] \qquad (12)$$

to be the error in the kernel estimate at the kernel parameter $w$. We give our argument by duality, noting that the lemma is equivalent to proving

$$\mathbb{P}\left(\sup_{Q \in \mathcal{P}} \left| \int \widehat{e}_n(w) dQ(w) \right| \geq t \right) \leq \sqrt{2} \exp\left(-\frac{nt^2}{16(\rho+1)}\right).$$

Before continuing, we note the following useful result, whose proof we provide in Section B.3.

**Lemma 6.** *For each fixed $w$, the random variable $\widehat{e}_n(w)$ is mean-zero and $\frac{4}{n}$-sub-Gaussian.*

To simplify the algebra, we work with a scaled version of the $f$-divergence: $f(t) = \frac{1}{k}(t^k - 1)$, so the equivalent constraint sets are $\mathcal{P} := \left\{Q : D_f(Q\|P_0) \leq \frac{\rho}{k}\right\}$ and $\mathcal{P}_{N_w} := \left\{q : D_f(q\|\mathbf{1}/N_w) \leq \frac{\rho}{k}\right\}$. In this rescaled form, the convex conjugate of $f(t)$ is $f^*(s) = \frac{1}{k_*}[s]_+^{k_*} + \frac{1}{k}$, where we recall the definition that $\frac{1}{k} + \frac{1}{k_*} = 1$.

Using Lemma 4, we obtain

$$
\begin{aligned}
\sup_{Q \in \mathcal{P}} \left| \int \widehat{e}_n(w) dQ(w) \right| &\leq \sup_{Q \in \mathcal{P}} \int |\widehat{e}_n(w)| \, dQ(w) \\
&\leq \inf_{\lambda \geq 0} \left\{ \frac{1}{k_*} \mathbb{E}_{P_0}[|\widehat{e}_n(W)|^{k_*}] \lambda^{1-k_*} + \frac{\rho+1}{k}\lambda \right\} \\
&= (\rho+1)^{\frac{1}{k}} \mathbb{E}_{P_0}[|\widehat{e}_n(W)|^{k_*}]^{1/k_*} \\
&\leq \sqrt{\rho+1} \mathbb{E}_{P_0}[\widehat{e}_n(W)^2]^{\frac{1}{2}},
\end{aligned}
$$

where the second inequality follows by using $\eta = 0$ in Lemma 4 and the last inequality follows from the fact that $k \geq 2$ and $k_* \leq 2$. The expectation $\mathbb{E}_{P_0}$ is with respect to the variable $W$ for a fixed $\widehat{e}_n$. We now see that to prove the theorem, it suffices to show that

$$\mathbb{P}\left(\int \widehat{e}_n(w)^2 dP_0(w) \geq \frac{t^2}{\rho+1}\right) \leq \sqrt{2} \exp\left(-\frac{nt^2}{16(\rho+1)}\right).$$

By Lemma 6, $\widehat{e}_n$ is $4/n$-sub-Gaussian, whence $\mathbb{E}\left[\exp\left(\lambda \widehat{e}_n(w)^2\right)\right] \leq \exp\left(-\frac{1}{2}\log\left(1 - \frac{8\lambda}{n}\right)\right)$ for $\lambda \leq \frac{n}{8}$ (recall inequality (10) above). Integrating over $w$, we find that for any distribution $P_0$ we have by the Chernoff bound technique that for $\lambda \leq \frac{n}{8}$,

$$
\begin{aligned}
\mathbb{P}\left(\int \widehat{e}_n(w)^2 dP_0(w) \geq \frac{t^2}{\rho+1}\right) &\leq \mathbb{E}\left[\exp\left(\lambda \int \widehat{e}_n(w)^2 dP(w)\right)\right] \exp\left(-\lambda \frac{t^2}{\rho+1}\right) \\
&\leq \int \mathbb{E}\left[\exp\left(\lambda \widehat{e}_n(w)^2\right)\right] dP(w) \exp\left(-\lambda \frac{t^2}{\rho+1}\right) \\
&\leq \exp\left(-\frac{1}{2}\log\left(1 - \frac{8\lambda}{n}\right)\right) \exp\left(-\lambda \frac{t^2}{\rho+1}\right).
\end{aligned}
$$

Note that $-\log(1-t) \leq t \log 4$ for $t \leq \frac{1}{2}$, and take $\lambda = n/16$ to get the result.

## A.2 Proof of Lemma 2

Let $F : \mathcal{W} \to [-\|F\|_\infty, \|F\|_\infty]$ be a function of the random $W$. In our setting, this map is equal to

$$F(w) = \frac{1}{n(n-1)} \sum_{i \neq j} S_{ij} \phi(x^i, w) \phi(x^j, w),$$

where we treat the $S_{ij}$ and $x^i$ as fixed and work conditionally; that is, only $W$ is random. We consider the convergence of

$$\sup_{Q \in \mathcal{P}_{N_w}} \mathbb{E}_Q[F(W)] \quad \text{to} \quad \sup_{Q \in \mathcal{P}} \mathbb{E}_Q[F(W)].$$

In the sequel, we suppress dependence on $W$ for notational convenience, and for a sample $W_1, \ldots, W_{N_w}$ of random vectors $W_k$, we let

$$F_k = \frac{1}{n(n-1)} \sum_{i \neq j} S_{ij} \phi(x^i, W_k) \phi(x^j, W_k)$$

for shorthand, so that the $F_k$ are bounded indepenent random variables.

Treating $F = (F_1, \ldots, F_{N_w})$ as a vector, the mapping $F \mapsto \sup_{Q \in \mathcal{P}_{N_w}} \mathbb{E}_Q[F]$ is a Lipschitz convex function of independent bounded random variables. Indeed, letting $q \in \mathbb{R}_+^{N_w}$ be the empirical probability mass function associated with $Q \in \mathcal{P}_{N_w}$ and recalling that $\|x\|_2 \leq n^{\frac{k-2}{2k}} \|x\|_k$ for $x \in \mathbb{R}^n$ and $k \geq 2$, we have $\frac{1}{N_w} \sum_{i=1}^{N_w} (N_w q_i)^k \leq \rho + 1$, which is equivalent to

$$\|q\|_2 \leq N_w^{\frac{k-2}{2k}} \|q\|_k \leq N_w^{\frac{k-2}{2k}} (\rho+1)^{\frac{1}{k}} N_w^{1/k-1} = (\rho+1)^{\frac{1}{k}} N_w^{-\frac{1}{2}}. \tag{13}$$

That is, the function $(F_1, \ldots, F_{N_w}) \mapsto \sup_{Q \in \mathcal{P}_{N_w}} \mathbb{E}_Q[F]$ is an $L_{N_w} = \sqrt{\rho+1}/\sqrt{N_w}$-Lipschitz and convex function of bounded random variables. Using Samson's sub-Gaussian concentration inequality [26] for Lipschitz convex functions of bounded random variables, we have with probability at least $1 - \delta$ that

$$\sup_{Q \in \mathcal{P}_{N_w}} \mathbb{E}_Q[F] \in \mathbb{E}\left[ \sup_{Q \in \mathcal{P}_{N_w}} \mathbb{E}_Q[F] \right] \pm 2\sqrt{2} \|F\|_\infty \sqrt{\frac{(1+\rho)\log \frac{2}{\delta}}{N_w}}. \tag{14}$$

By the containment (14), we need consider only the convergence of the expectation

$$\mathbb{E}\left[ \sup_{Q \in \mathcal{P}_{N_w}} \mathbb{E}_Q[F] \right] \quad \text{to} \quad \sup_{Q \in \mathcal{P}} \mathbb{E}_Q[F].$$

But of course, this convergence is described precisely by Lemma 5. Thus, combining Lemma 5 with containment (14) gives

$$\left| \sup_{Q \in \mathcal{P}_{N_w}} \mathbb{E}_Q[F] - \sup_{Q \in \mathcal{P}} \mathbb{E}_Q[F] \right| \leq 4 C_\rho \|F\|_\infty \sqrt{\frac{\log(2N_w)}{N_w}} + 2\sqrt{2} \|F\|_\infty \sqrt{\frac{(1+\rho)\log \frac{2}{\delta}}{N_w}}$$

Now, since $\|F\|_\infty = 1$ we can simplify this to get the result.

## A.3 Proof of Theorem 1

We can write

$$\left| T(\widehat{Q}_w) - \sup_{Q \in \mathcal{P}} T(Q) \right| \leq \left| \sup_{Q \in \mathcal{P}} T(Q) - \sup_{Q \in \mathcal{P}} \widehat{T}(Q) \right| + \left| \sup_{Q \in \mathcal{P}} \widehat{T}(Q) - \widehat{T}(\widehat{Q}_w) \right| + \left| \widehat{T}(\widehat{Q}_w) - T(\widehat{Q}_w) \right|$$

$$\leq \sup_{Q \in \mathcal{P}} \left| T(Q) - \widehat{T}(Q) \right| + \left| \sup_{Q \in \mathcal{P}} \widehat{T}(Q) - \widehat{T}(\widehat{Q}_w) \right| + \sup_{Q \in \mathcal{P}_{N_w}} \left| \widehat{T}(Q) - T(Q) \right|$$

Now apply Lemma 1 to the first and third terms, apply Lemma 2 to the second term, and use a union bound to get the result.

## A.4 Proof of Lemma 3

We define define the "dual" representation of the feature matrix: let $\Psi = \Phi^T = [\psi^1 \; \cdots \; \psi^{N_w}]$, with columns given by $\psi^m := [\phi(x^1, w^m) \; \cdots \; \phi(x^n, w^m)]^T \in \mathbb{R}^n$. Mimicking the proof of Proposition 1 of [8], we have

$$\mathcal{R}_n(\mathcal{F}_{N_w}) = \frac{B}{n} \mathbb{E}\left[ \sup_{q \in \mathcal{P}_{N_w}} \sqrt{\sigma^T \left( \sum_{k=1}^{N_w} q_k \psi^k (\psi^k)^T \right) \sigma} \right], \tag{15}$$

where $\sigma_i \in \{-1, 1\}$ are iid. Rademacher variables. By the bound (13), the containment $q \in \mathcal{P}_{N_w}$ implies the bound $\|q\|_2 \leq \sqrt{(1+\rho)/N_w}$, so

$$\mathcal{R}_n(\mathcal{F}_{N_w}) \leq \frac{B}{n} \mathbb{E}\left[ \sqrt{\sqrt{\frac{1+\rho}{N_w}} \sum_{k=1}^{N_w} \frac{(\sigma^T \psi^k)^4}{\sqrt{\sum_{a=1}^{N_w} (\sigma^T \psi^a)^4}}} \right]$$

$$= \frac{B}{n} \mathbb{E}\left[ \left( \frac{1+\rho}{N_w} \sum_{k=1}^{N_w} (\sigma^T \psi^k)^4 \right)^{\frac{1}{4}} \right]$$

$$\leq \frac{B}{n} \left( \mathbb{E}\left[ \frac{1+\rho}{N_w} \sum_{k=1}^{N_w} (\sigma^T \psi^k)^4 \right] \right)^{\frac{1}{4}},$$

where the first inequality follows from the Cauchy-Schwarz inequality and the second inequality is Jensen's inequality. As $\psi_i \in [-1, 1]$, we have

$$\mathbb{E}\left[ (\sigma^T \psi)^4 \right] \leq \mathbb{E}\left[ \left( \sum_{i=1}^n \sigma_i \right)^4 \right]$$

$$= 3n^2 - 2n \leq 3n^2.$$

Then

$$\mathcal{R}_n(\mathcal{F}_{N_w}) \leq \frac{B}{n} \left( 3(1+\rho)n^2 \right)^{\frac{1}{4}} \leq B\sqrt{\frac{2(1+\rho)}{n}}$$

as desired.

# B Technical lemmas

## B.1 Proof of Lemma 4

Let $L \geq 0$ satisfy $L(w) = dQ(w)/dP(w)$, so that $L$ is the likelihood ratio between $Q$ and $P$. Then we have

$$\sup_{Q:D_f(Q\|P)\leq\rho} \int g(w)dQ(w) = \sup_{\int f(L)dP\leq\rho, \mathbb{E}_P[L]=1} \int g(w)L(w)dP(w)$$

$$= \sup_{L\geq 0} \inf_{\lambda\geq 0,\eta} \left\{ \int g(w)L(w)dP(w) - \lambda \left( \int f(L(w))dP(w) - \rho \right) - \eta \left( \int L(w)dP(w) - 1 \right) \right\}$$

$$= \inf_{\lambda\geq 0,\eta} \sup_{L\geq 0} \left\{ \int g(w)L(w)dP(w) - \lambda \left( \int f(L(w))dP(w) - \rho \right) - \eta \left( \int L(w)dP(w) - 1 \right) \right\},$$

where we have used that strong duality obtains because the problem is strictly feasible in its non-linear constraints (take $L \equiv 1$), so that the extended Slater condition holds [22, Theorem 8.6.1 and Problem 8.7]. Noting that $L$ is simply a positive (but otherwise arbitrary) function, we obtain

$$\sup_{Q:D_f(Q\|P)\leq\rho} \int g(w)dQ(w) = \inf_{\lambda\geq 0,\eta} \int \sup_{\ell\geq 0} \left\{ (g(w) - \eta)\ell - \lambda f(\ell) \right\} dP(w) + \lambda\rho + \eta$$

$$= \inf_{\lambda\geq 0,\eta} \int \lambda f^* \left( \frac{g(w) - \eta}{\lambda} \right) dP(w) + \eta + \rho\lambda.$$

Here we have used that $f^*(s) = \sup_{t \geq 0}\{st - f(t)\}$ is the conjugate of $f$ and that $\lambda \geq 0$, so that we may take divide and multiply by $\lambda$ in the supremum calculation.

## B.2 Proof of Lemma 5

We remark that the upper bound in the lemma is immediate from the argument for inequality (11). Thus we focus only on the lower bound claimed in the lemma.

Before beginning the proof proper, we state a useful lemma lower bounding expectations of various moments of random variables. (See Section B.4 for a proof.)

**Lemma 7.** *Let $Z \geq 0$, $Z \not\equiv 0$ be a random variable with finite $2p$-th moment for $1 \leq p \leq \infty$. Then we have the following inequalities:*

$$
\mathbb{E}\left[\left(\frac{1}{n}\sum_{i=1}^{n} Z_i^p\right)^{\frac{1}{p}}\right]
$$

$$
\geq \|Z\|_p - \begin{cases} \frac{p-1}{p}\sqrt{\frac{2}{n}}\sqrt{\mathrm{Var}(Z^p/\mathbb{E}[Z^p])}\|Z\|_2, & \text{if } p \leq 2 \\ 2\min\left(\frac{p-1}{p}\sqrt{\frac{1}{n}}\sqrt{\mathrm{Var}(Z^p/\mathbb{E}[Z^p])}\|Z\|_p, \frac{1}{n}\left(\frac{p-1}{p}\right)^2 \frac{\mathrm{Var}(Z^p)}{\|Z\|_p^{2p-1}}\right) & \text{if } p \geq 2. \end{cases} \quad (16\text{a})
$$

*and if $\|Z\|_\infty \leq C$, then*

$$
\mathbb{E}\left[\left(\frac{1}{n}\sum_{i=1}^{n} Z_i^p\right)^{\frac{1}{p}}\right] \geq \|Z\|_p - \begin{cases} C\frac{p-1}{p}\sqrt{\frac{2}{n}}, & \text{if } p \leq 2 \\ 2C\left(\frac{1}{n}\right)^{\frac{1}{p}} & \text{if } p > 2 \end{cases} \quad (16\text{b})
$$

For convenience in the proof to follow, we define the shorthand

$$
S_{N_w}(\eta) := (1+\rho)^{1/k}\left(\frac{1}{N_w}\sum_{i=1}^{N_w}[Z_i - \eta]_+^{k_*}\right)^{\frac{1}{k_*}} + \eta.
$$

We also rescale $\rho$ to $\rho/k$ for algebraic convenience. For the function $f(t) = \frac{1}{k}(t^k - 1)$, we have $f^*(s) = \frac{1}{k_*}[s]_+^{k_*} + \frac{1}{k}$, so that the duality result in Lemma 4 shows that (after taking an infimum over $\lambda \geq 0$)

$$
\sup_{Q \in \mathcal{P}_{N_w}}\mathbb{E}_Q[Z] = \inf_\eta\left\{(1+\rho)^{1/k}\left(\frac{1}{N_w}\sum_{i=1}^{N_w}[Z_i - \eta]_+^{k_*}\right)^{\frac{1}{k_*}} + \eta\right\}.
$$

Because $|Z_i| \leq M$ for all $i$, we claim that any $\eta$ minimizing the preceding expression must satisfy

$$
\eta \in \left[-\frac{1 + (1+\rho)^{\frac{1}{k_*}}}{(1+\rho)^{\frac{1}{k_*}} - 1}, 1\right] \cdot M. \quad (17)
$$

Indeed, it is clear that $\eta \leq M$, because otherwise we would have $S_{N_w}(\eta) > M \geq \inf_\eta S_{N_w}(\eta)$. The lower bound on $\eta$ is somewhat less trivial. Let $\eta = -cM$ for some $c > 1$. Taking derivatives of the objective $S_{N_w}(\eta)$ with respect to $\eta$, we have

$$
S'_{N_w}(\eta) = 1 - (1+\rho)^{1/k}\frac{\frac{1}{N_w}\sum_{i=1}^{N_w}[Z_i - \eta]_+^{k_*-1}}{\left(\frac{1}{N_w}\sum_{i=1}^{N_w}[Z_i - \eta]_+^{k_*}\right)^{1-\frac{1}{k_*}}} \leq 1 - (1+\rho)^{1/k}\left(\frac{(c-1)M}{(c+1)M}\right)^{k_*-1}
$$

$$
= 1 - (1+\rho)^{1/k}\left(\frac{c-1}{c+1}\right)^{k_*-1}.
$$

Defining the constant $c_{\rho,k} := \frac{(1+\rho)^{\frac{1}{k_*}} + 1}{(1+\rho)^{\frac{1}{k_*}} - 1}$, we see that for any $c > c_{\rho,k}$, the preceding display is negative, so we must have $\eta \geq -c_{\rho,k}M$ (since the derivative is 0 at optimality). For the remainder of the proof, we thus define the interval

$$
U := [-Mc_{\rho,k}, M], \quad c_{\rho,k} = \frac{(1+\rho)^{\frac{1}{k_*}} + 1}{(1+\rho)^{\frac{1}{k_*}} - 1},
$$

and we assume w.l.o.g. that $\eta \in U$.

Again applying the duality result of Lemma 4, we have that

$$\mathbb{E}\left[\sup_{Q \in \mathcal{P}_{N_w}} \mathbb{E}_Q[Z]\right] = \mathbb{E}\left[\inf_{\eta \in U} S_{N_w}(\eta)\right] = \mathbb{E}\left[\inf_{\eta \in U}\{S_{N_w}(\eta) - \mathbb{E}[S_{N_w}(\eta)] + \mathbb{E}[S_{N_w}(\eta)]\}\right]$$

$$\geq \inf_{\eta \in U} \mathbb{E}[S_{N_w}(\eta)] - \mathbb{E}\left[\sup_{\eta \in U} |S_{N_w}(\eta) - \mathbb{E}[S_{N_w}(\eta)]|\right]. \tag{18}$$

To bound the first term in expression (18), note that $[Z - \eta]_+ \in [0, 1 + c_{\rho,k}]M$ and $(1 + \rho)^{1/k}(1 + c_{\rho,k}) = C_{\rho,k}$. Thus, by Lemma 7 we obtain that

$$\mathbb{E}[S_{N_w}(\eta)] \geq (1 + \rho)^{1/k}\mathbb{E}\left[[Z - \eta]_+^{k_*}\right]^{1/k_*} + \eta - C_{\rho,k}M\frac{k_* - 1}{k_*}\sqrt{\frac{2}{N_w}}.$$

Using that $\frac{k_* - 1}{k_*} = \frac{1}{k}$, taking the infimum over $\eta$ on the right hand side and using duality yields

$$\inf_{\eta} \mathbb{E}[S_{N_w}(\eta)] \geq \sup_{Q \in \mathcal{P}} \mathbb{E}_Q[Z] - C_{\rho,k}\frac{M}{k}\sqrt{\frac{2}{N_w}}.$$

To bound the second term in expression (18), we use concentration results for Lipschitz functions. First, the function $\eta \mapsto S_{N_w}(\eta)$ is $\sqrt{1 + \rho}$-Lipschitz in $\eta$. To see this, note that for $1 \leq k^\star \leq 2$ and $X \geq 0$, by Jensen's inequality,

$$\frac{\mathbb{E}[X^{k^\star - 1}]}{(\mathbb{E}[X^{k^\star}])^{1 - 1/k^\star}} \leq \frac{\mathbb{E}[X]^{k^\star - 1}}{(\mathbb{E}[X^{k^\star}])^{1 - 1/k^\star}} \leq \frac{\mathbb{E}[X]^{k^\star - 1}}{\mathbb{E}[X]^{k^\star - 1}} = 1,$$

so $S'_{N_w}(\eta) \in [1 - (1 + \rho)^{\frac{1}{k}}, 1]$ and therefore $S_{N_w}$ is $(1 + \rho)^{1/k}$-Lipschitz in $\eta$. Furthermore, the mapping $T : z \mapsto (1 + \rho)^{\frac{1}{k}}\left(\frac{1}{N_w}\sum_{i=1}^{N_w}[z_i - \eta]_+^{k_*}\right)^{\frac{1}{k_*}}$ for $z \in \mathbb{R}^{N_w}$ is convex and $(1 + \rho)^{\frac{1}{k}}/\sqrt{N_w}$-Lipschitz. This is verified by the following:

$$|T(z) - T(z')| \leq (1 + \rho)^{1/k}\left|\left(\frac{1}{N_w}\sum_{i=1}^{N_w}\left|[z_i - \eta]_+ - [z'_i - \eta]_+\right|^{k_*}\right)^{\frac{1}{k_*}}\right|$$

$$\leq \frac{(1 + \rho)^{1/k}}{N_w^{1/k_*}}\left|\left(\sum_{i=1}^{N_w}|z_i - z'_i|^{k_*}\right)^{\frac{1}{k_*}}\right|$$

$$\leq \frac{(1 + \rho)^{1/k}}{\sqrt{N_w}}\|z - z'\|_2,$$

where the first inequality is Minkowski's inequality and the third inequality follows from the fact that for any vector $x \in \mathbb{R}^n$, we have $\|x\|_p \leq n^{\frac{2-p}{2p}}\|x\|_2$ for $p \in [1, 2]$, where these denote the usual vector norms. Thus, the mapping $Z \mapsto S_{N_w}(\eta)$ is $(1 + \rho)^{1/k}/\sqrt{N_w}$-Lipschitz continuous with respect to the $\ell_2$-norm on $Z$. Again applying Samson's sub-Gaussian concentration result for convex Lipschitz functions, we have

$$\mathbb{P}\left(|S_{N_w}(\eta) - \mathbb{E}[S_{N_w}(\eta)]| \geq \delta\right) \leq 2\exp\left(-\frac{N_w\delta^2}{2C_{\rho,k}^2 M^2}\right)$$

for any fixed $\eta \in \mathbb{R}$ and any $\delta \geq 0$. Now, let $\mathcal{N}(U, \epsilon) = \{\eta_1, \ldots, \eta_{N(U,\epsilon)}\}$ be an $\epsilon$ cover of the set $U$, which we may take to have size at most $N(U, \epsilon) \leq M(1 + c_{\rho,k})\frac{1}{\epsilon}$. Then we have

$$\sup_{\eta \in U} |S_{N_w}(\eta) - \mathbb{E}[S_{N_w}(\eta)]| \leq \max_{i \in \mathcal{N}(U,\epsilon)} |S_{N_w}(\eta_i) - \mathbb{E}[S_{N_w}(\eta_i)]| + \epsilon(1 + \rho)^{1/k}.$$

Using the fact that $\mathbb{E}[\max_{i \leq n} |X_i|] \leq \sqrt{2\sigma^2 \log(2n)}$ for $X_i$ all $\sigma^2$-sub-Gaussian, we have

$$\mathbb{E}\left[\max_{i \in \mathcal{N}(U,\epsilon)} |S_{N_w}(\eta_i) - \mathbb{E}[S_{N_w}(\eta_i)]|\right] \leq C_{\rho,k}\sqrt{2\frac{M^2}{N_w}\log 2N(U, \epsilon)}.$$

Taking $\epsilon = M(1 + c_{\rho,k})/N_w$ gives that

$$\mathbb{E}\left[\sup_{\eta \in U} |S_{N_w}(\eta) - \mathbb{E}[S_{N_w}(\eta)]|\right] \leq \sqrt{2}MC_{\rho,k}\sqrt{\frac{1}{N_w}\log(2N_w)} + \frac{C_{\rho,k}M}{N_w}.$$

Then, in total we have (using $C_\rho \geq C_{\rho,k}$, $k \geq 2$, and $N_w \geq 1$),

$$\mathbb{E}\left[\sup_{Q \in \mathcal{P}_{N_w}} \mathbb{E}_Q[Z]\right] \geq \sup_{Q \in \mathcal{P}} \mathbb{E}_Q[Z] - \frac{C_\rho M \sqrt{2}}{\sqrt{N_w}}\left(\frac{1}{k} + \sqrt{\log(2N_w)} + \frac{1}{\sqrt{2N_w}}\right)$$

$$\geq \sup_{Q \in \mathcal{P}} \mathbb{E}_Q[Z] - 4C_\rho M \sqrt{\frac{\log(2N_w)}{N_w}}.$$

This gives the desired result of the lemma.

### B.3 Proof of Lemma 6

The result follows from bounded differences. First, we let

$$\widehat{e}'_n(w) = \frac{1}{n(n-1)} \sum_{i \neq j} S'_{ij}\phi(x'_i; w)\phi(x'_j; w) - \mathbb{E}[S(X, X')\phi(X, w)\phi(X', w)],$$

where we assume $d_{\text{ham}}(x_{1:n}, x'_{1:n}) \leq 1$ and $S_{ij} = S'_{ij}$ except for those pairs $(i, j)$ such that $x'_i \neq x_i$ or $x_j \neq x'_j$. Assuming (without loss of generality by symmetry) that $x_{2:n} = x'_{2:n}$, we have

$$|\widehat{e}_n(w) - \widehat{e}'_n(w)| \leq \frac{1}{n(n-1)} \sum_{j>1} |S_{1j}\phi(x_1; w)\phi(x_j; w) - S'_{1j}\phi(x'_1; w)\phi(x_j; w)|$$

$$+ \frac{1}{n(n-1)} \sum_{i>1} |S_{i1}\phi(x_i; w)\phi(x_1; w) - S'_{i1}\phi(x'_i; w)\phi(x'_1; w)|$$

$$\leq \frac{2(n-1)}{n(n-1)} + \frac{2(n-1)}{n(n-1)} = \frac{4}{n},$$

where in the last line we have used that $\max\{\|\phi\|_\infty, \|S\|_\infty\} \leq 1$. In particular, $\widehat{e}_n(w)$ has bounded differences and is mean zero, so that the usual construction with Doob martingales yields

$$\mathbb{E}\left[\exp(\lambda \widehat{e}_n(w))\right] \leq \exp\left(\frac{16\lambda^2}{8n^2}\right)^n = \exp\left(\frac{2\lambda^2}{n}\right).$$

This is the desired result.

### B.4 Proof of Lemma 7

For $a > 0$, we have

$$\inf_{\lambda \geq 0} \left\{\frac{a^p}{p\lambda^{p-1}} + \lambda\frac{p-1}{p}\right\} = a,$$

(with $\lambda = a$ attaining the infimum), and taking derivatives yields

$$\frac{a^p}{p\lambda^{p-1}} + \lambda\frac{p-1}{p} \geq \frac{a^p}{p\lambda_1^{p-1}} + \lambda_1\frac{p-1}{p} + \frac{p-1}{p}\left(1 - \frac{a^p}{\lambda_1^p}\right)(\lambda - \lambda_1).$$

Using this in the moment expectation, by setting $\lambda_n = \sqrt[p]{\frac{1}{n}\sum_{i=1}^n Z_i^p}$, we have for any $\lambda \geq 0$ that

$$\mathbb{E}\left[\left(\frac{1}{n}\sum_{i=1}^n Z_i^p\right)^{\frac{1}{p}}\right] = \mathbb{E}\left[\frac{\sum_{i=1}^n Z_i^p}{pn\lambda_n^{p-1}} + \lambda_n\frac{p-1}{p}\right]$$

$$\geq \mathbb{E}\left[\frac{\sum_{i=1}^n Z_i^p}{pn\lambda^{p-1}} + \lambda\frac{p-1}{p}\right] + \frac{p-1}{p}\mathbb{E}\left[\left(1 - \frac{\sum_{i=1}^n Z_i^p}{n\lambda^p}\right)(\lambda_n - \lambda)\right].$$

Now we take $\lambda = \|Z\|_p$, and we apply the Cauchy-Schwarz inequality to obtain

$$\mathbb{E}\left[\left(\frac{1}{n}\sum_{i=1}^{n}Z_i^p\right)^{\frac{1}{p}}\right] \geq \|Z\|_p - \frac{p-1}{p}\mathbb{E}\left[\left(1 - \frac{\frac{1}{n}\sum_{i=1}^{n}Z_i^p}{\|Z\|_p^p}\right)^2\right]^{\frac{1}{2}}\mathbb{E}\left[\left(\left(\frac{1}{n}\sum_{i=1}^{n}Z_i^p\right)^{\frac{1}{p}} - \|Z\|_p\right)^2\right]^{\frac{1}{2}}$$

$$= \|Z\|_p - \frac{p-1}{p\sqrt{n}}\sqrt{\mathrm{Var}(Z^p/\mathbb{E}[Z^p])}\mathbb{E}\left[\left(\left(\frac{1}{n}\sum_{i=1}^{n}Z_i^p\right)^{\frac{1}{p}} - \mathbb{E}[Z^p]^{\frac{1}{p}}\right)^2\right]^{\frac{1}{2}} \qquad (19)$$

$$\geq \|Z\|_p - \frac{p-1}{p\sqrt{n}}\sqrt{\mathrm{Var}(Z^p/\mathbb{E}[Z^p])}\mathbb{E}\left[\left(\frac{1}{n}\sum_{i=1}^{n}Z_i^p\right)^{\frac{2}{p}} + \mathbb{E}[Z^p]^{\frac{2}{p}}\right]^{\frac{1}{2}}.$$

Now, for $p \leq 2$, we have

$$\mathbb{E}\left[\left(\frac{1}{n}\sum_{i=1}^{n}Z_i^p\right)^{\frac{1}{p}}\right] \geq \|Z\|_p - \frac{p-1}{p}\sqrt{\frac{2}{n}}\sqrt{\mathrm{Var}(Z^p/\mathbb{E}[Z^p])}\|Z\|_2,$$

by Jensen, or equivalently, the fact that the norm is non-decreasing in $p$. For $p \geq 2$, we have by the triangle inequality applied to expression (19), followed by an application of Jensen's inequality (using that $\mathbb{E}[Y^{2/p}] \leq \mathbb{E}[Y]^{2/p}$ for $p \geq 2$),

$$\mathbb{E}\left[\left(\frac{1}{n}\sum_{i=1}^{n}Z_i^p\right)^{\frac{1}{p}}\right] \geq \|Z\|_p - 2\frac{p-1}{p}\sqrt{\frac{1}{n}}\sqrt{\mathrm{Var}(Z^p/\mathbb{E}[Z^p])}\|Z\|_p,$$

Now, we can make this tighter (for $p \geq 2$):

$$\mathbb{E}\left[\left(\left(\frac{1}{n}\sum_{i=1}^{n}Z_i^p\right)^{\frac{1}{p}} - \mathbb{E}[Z^p]^{\frac{1}{p}}\right)^2\right] = \mathbb{E}\left[\left(\frac{1}{n}\sum_{i=1}^{n}Z_i^p\right)^{\frac{2}{p}}\right] + \|Z\|_p^2 - 2\|Z\|_p\mathbb{E}\left[\left(\frac{1}{n}\sum_{i=1}^{n}Z_i^p\right)^{\frac{1}{p}}\right]$$

$$\leq 2\|Z\|_p^2 - 2\|Z\|_p\mathbb{E}\left[\left(\frac{1}{n}\sum_{i=1}^{n}Z_i^p\right)^{\frac{1}{p}}\right]$$

$$\leq 2\frac{p-1}{p}\frac{2}{\sqrt{n}}\sqrt{\mathrm{Var}(Z^p/\mathbb{E}[Z^p])}\|Z\|_p^2.$$

Further, we can recurse this argument. Let

$$Y := \mathbb{E}\left[\left(\frac{1}{n}\sum_{i=1}^{n}Z_i^p\right)^{\frac{1}{p}}\right]$$

$$A := \|Z\|_p$$

$$B := \frac{p-1}{p}\sqrt{\frac{1}{n}}\sqrt{\mathrm{Var}(Z^p/\mathbb{E}[Z^p])},$$

$$C := \mathbb{E}\left[\left(\left(\frac{1}{n}\sum_{i=1}^{n}Z_i^p\right)^{\frac{1}{p}} - \mathbb{E}[Z^p]^{\frac{1}{p}}\right)^2\right].$$

Then, we have three primary relationships $r : Y \geq A - BC^{\frac{1}{2}}$, $s_0 : C \leq 2A^2 - 2AY$, and $t_0 : Y \geq A - 2AB$. Recursion works as follows: for $i \geq 0$, we plug $t_i$ into $s_0$ to yield a tighter inequality $s_{i+1}$ for $C$, which in turn plugs in to $r$ to yield a tighter inequality $t_{i+1}$ for $Y$. In this way, we have the relations $s_i : C \leq 4A^2 B^{a_{i-1}}$ for $i \geq 1$, and $t_i : Y \geq A - 2AB^{a_i}$ for $i \geq 0$, where

$a_i = 2 - 2^{-i}$. Taking $i \to \infty$, we have $Y \geq A - 2AB^2$, or

$$\mathbb{E}\left[\left(\frac{1}{n}\sum_{i=1}^{n} Z_i^p\right)^{\frac{1}{p}}\right] \geq \|Z\|_p - 2\|Z\|_p \left(\frac{p-1}{p}\right)^2 \frac{\operatorname{Var}(Z^p/\mathbb{E}[Z^p])}{n}$$

$$= \|Z\|_p - \frac{2}{n}\left(\frac{p-1}{p}\right)^2 \frac{\operatorname{Var}(Z^p)}{\|Z\|_p^{2p-1}}$$

Thus, we have

$$\mathbb{E}\left[\left(\frac{1}{n}\sum_{i=1}^{n} Z_i^p\right)^{\frac{1}{p}}\right] \geq \|Z\|_p - \begin{cases} \frac{p-1}{p}\sqrt{\frac{2}{n}}\sqrt{\operatorname{Var}(Z^p/\mathbb{E}[Z^p])}\|Z\|_2, & \text{if } p \leq 2 \\ 2\min\left(\frac{p-1}{p}\sqrt{\frac{1}{n}}\sqrt{\operatorname{Var}(Z^p/\mathbb{E}[Z^p])}\|Z\|_p, \frac{1}{n}\left(\frac{p-1}{p}\right)^2 \frac{\operatorname{Var}(Z^p)}{\|Z\|_p^{2p-1}}\right) & \text{if } p \geq 2 \end{cases}$$

In the case that we have the unifom bound $\|Z\|_\infty \leq C$, we can get tighter guarantees. To that end, we state a simple lemma.

**Lemma 8.** *For any random variable $X \geq 0$ and $a \in [1,2]$, we have*

$$\mathbb{E}[X^{ak}] \leq \mathbb{E}[X^k]^{2-a}\mathbb{E}[X^{2k}]^{a-1}$$

**Proof** For $c \in [0,1]$, $1/p + 1/q = 1$ and $A \geq 0$, we have by Holder's inequality,

$$\mathbb{E}[A] = \mathbb{E}[A^c A^{1-c}] \leq \mathbb{E}[A^{pc}]^{1/p}\mathbb{E}[A^{q(1-c)}]^{1/q}$$

Now take $A := X^{ak}$, $1/p = 2 - a$, $1/q = a - 1$, and $c = \frac{2}{a} - 1$. $\qquad\square$

First, note that $\mathbb{E}[Z^{2p}] \leq C^p \mathbb{E}[Z^p]$. For $1 \leq p \leq 2$, we can take $a = 2/p$ in Lemma 8, so that we have

$$E[Z^2] \leq \mathbb{E}[Z^p]^{2-\frac{2}{p}}\mathbb{E}[Z^{2p}]^{\frac{2}{p}-1} \leq \|Z\|_p^p C^{2-p}.$$

Now, we can plug these into the expression above (using $\operatorname{Var} Z^p \leq \mathbb{E}[Z^{2p}] \leq C^p\|Z\|_p^p$):

$$\mathbb{E}\left[\left(\frac{1}{n}\sum_{i=1}^{n} Z_i^p\right)^{\frac{1}{p}}\right] \geq \|Z\|_p - \begin{cases} C\frac{p-1}{p}\sqrt{\frac{2}{n}}, & \text{if } p \leq 2 \\ 2\min\left(\frac{p-1}{p}\sqrt{\frac{1}{n}}\sqrt{\operatorname{Var}(Z^p/\mathbb{E}[Z^p])}\|Z\|_p, \frac{1}{n}\left(\frac{p-1}{p}\right)^2 \frac{\operatorname{Var}(Z^p)}{\|Z\|_p^{2p-1}}\right) & \text{if } p \geq 2 \end{cases}$$

In fact, we can give a somewhat sharper result by noting that $\mathbb{E}[(\frac{1}{n}\sum_{i=1}^{n} Z_i^p)^{1/p}] \geq 0$, and similarly, $\|Z\|_p \geq 0$. For shorthand, let $D = (\frac{p-1}{p})^2 C^p$. Then using that $\operatorname{Var}(Z^p/\mathbb{E}[Z^p]) = \operatorname{Var}(Z^p)/\|Z\|_p^{2p} \leq \mathbb{E}[Z^{2p}]/\|Z\|_p^{2p} \leq C^p/\|Z\|_p^p$, the preceding inequality, in the case that $p \geq 2$, implies

$$\mathbb{E}\left[\left(\frac{1}{n}\sum_{i=1}^{n} Z_i^p\right)^{\frac{1}{p}}\right] \geq \|Z\|_p - 2\min\left\{\sqrt{D/n}\|Z\|_p^{1-p/2}, (D/n)\|Z\|_p^{1-p}, \|Z\|_p/2\right\}$$

$$\geq \|Z\|_p - 2\min\left\{\sqrt{D/n}\|Z\|_p^{1-p/2}, (D/n)\|Z\|_p^{1-p}, \|Z\|_p\right\}.$$

But now, we note that

$$\min_{t \geq 0}\left\{\sqrt{\frac{D}{n}}t^{1-p/2}, \frac{D}{n}t^{1-p}, t\right\} = \begin{cases} t, & \text{if } t \leq (D/n)^{1/p} \\ \frac{D}{n}t^{1-p}, & \text{if } t > (D/n)^{1/p} \end{cases}$$

$$\leq (D/n)^{1/p}.$$

In particular, we have for $p \geq 2$ that

$$\mathbb{E}\left[\left(\frac{1}{n}\sum_{i=1}^{n} Z_i^p\right)^{\frac{1}{p}}\right] \geq \|Z\|_p - 2\left(\frac{1}{n}\left(\frac{p-1}{p}\right)^2 C^p\right)^{1/p} \geq \|Z\|_p - 2C\left(\frac{1}{n}\right)^{\frac{1}{p}}.$$

Finally, we note that the bound for $p \leq 2$ is tighter than the above expression for $p = 2$.

# C More experiments

We present further details of the experiments shown in Section 4 as well as experiments on more datasets and kernel-learning methods. Specifically, we also show experiments with the `ads`[5], `farm`[6], `mnist`[7], and `weight`[8] datasets. When training/test splits do not already exist, we split the dataset into 75% training and 25% test sets.

Table 3 shows parameters used in our method for each dataset. The last column indicates the size of the subset of the training data used to solve problem (4). We use subsets to increase the efficiency of our approach. Furthermore, we show $\rho/N_w$ simply because it is easier to work with this quantity rather than $\rho$: the value is chosen to balance fit with efficiency via cross validation. Very large $\rho$ yields extremely sparse $\widehat{q}$ and poor fit, whereas very small $\rho$ yields dense $\widehat{q}$ and long training times. We note that all values of $\rho$ are less than 1000. Finally, for ridge regression models, we choose the $l_2$ penalty term such that we may absorb the $\sqrt{\widehat{q_i}}$ factors into $\theta$.

Table 4 compares the accuracy of our approach (OK) with other methods: random features with 2 values for $D$, and two standard multiple-kernel-learning algorithms from [14]. Table 5 shows the (training + test) times of the same methods. Algorithm ABMKSVM(ratio) is a heuristic alignment-based kernel derived in problem (2) in [14] followed by an SVM. Algorithm MKSVM jointly optimizes kernel composition with empirical risk via problem (9) in [14]. For both of these methods, we consider optimizing the combination of a linear, second-order polynomial, and Gaussian kernel.

The two multiple-kernel-learning approaches require an extremely large amount of memory to build Gram matrices, so we train on subsets of data when necessary to avoid latencies introduced by swapping data from memory. For ABMKSVM(ratio) we train on $n = 17500$ for `adult` and `weight`, and $n = 10000$ for `reuters`. Similarly, we break up the test data for `reuters` into $n_{test} = 1000$ chunks, which accounts for the large amount of time taken for this dataset (training time was roughly $400s$). For MKSVM, we use a subset of size $n = 7500$ for all applicable datasets, and we use the same testing scheme as ABMKSVM(ratio) for `reuters` (training time for MKSVM was roughly $1000s$).

The performance of our method on all datasets is consistent: we improve the performance for random features at a given computational cost, and we are generally competitive with much costlier standard multiple-kernel-learning techniques. The `mnist` and `weight` datasets are slightly peculiar: both ABSVM(ratio) and MKSVM require many support vectors, indicating that the chosen kernels are poor for the task; this hypothesis is corroborated by the slightly worse performance of both our method and random features (the arc-cosine kernel is similar to polynomial and Guassian kernels). A large number of support vectors roughly translates to large $\mathrm{nnz}(\widehat{q})$, which can be achieved by increasing $N_w$ or decreasing $\rho$. We can also achieve better performance by increasing the subset of training data used in problem (4). Doing the latter two options yields comparable results for our method (Table 6). For the `mnist` models, we switch to ridge regression to enhance efficiency of the larger problem. The upshot of this analysis is that our method is most effective in regimes where standard multiple-kernel-learning techniques are intractable, that is, datasets with both large $n$ and $d$.

**Table 3:** Dataset parameters

| Dataset | $n,$ | $n_{test}$ | $d$ | Model | Base kernel | $\rho/N_w$ | $N_w$ | %$n$ in problem (4) |
|---|---|---|---|---|---|---|---|---|
| adult | 32561, | 16281 | 123 | Logistic | Gaussian | 0.0120 | 20000 | 50 |
| reuters | 23149, | 781265 | 47236 | Ridge | Linear | 0.0123 | 47236 | 100 |
| buzz | 105530, | 35177 | 77 | Ridge | Arc-cosine | 0.0145 | 2000 | 6.67 |
| ads | 2459, | 820 | 1554 | Ridge | Linear | 0.1000 | 1554 | 100 |
| farm | 3107, | 1036 | 54877 | Ridge | Linear | 0.0050 | 54877 | 100 |
| mnist17 | 13007, | 2163 | 784 | Logistic | Arc-cosine | 0.0300 | 20000 | 25 |
| mnist49 | 11791, | 1991 | 784 | Logistic | Arc-cosine | 0.0300 | 20000 | 25 |
| mnist56 | 11339, | 1850 | 784 | Logistic | Arc-cosine | 0.0300 | 20000 | 25 |
| weight | 29431, | 9811 | 53 | Ridge | Gaussian | 0.0020 | 20000 | 50 |

**Table 4:** Test misclassification error (%)

| Dataset | OK $D = \mathrm{nnz}(\widehat{q})$ | Random $D = \mathrm{nnz}(\widehat{q})$ | Random $D = 10\,\mathrm{nnz}(\widehat{q})$ | ABMKSVM(ratio) | MKSVM |
|---|---|---|---|---|---|
| adult | 15.54 | 17.51 | 16.08 | 15.44 | 16.79 |
| reuters | 9.27 | 46.49 | 23.69 | 9.09 | 10.13 |
| buzz | 4.92 | 8.68 | 4.16 | 3.48 | 3.54 |
| ads | 5.37 | 8.05 | 3.54 | 3.05 | 3.17 |
| farm | 11.58 | 23.36 | 14.58 | 10.81 | 10.23 |
| mnist17 | 3.24 | 4.44 | 1.76 | 0.51 | 0.97 |
| mnist49 | 6.53 | 21.55 | 4.02 | 1.10 | 1.26 |
| mnist56 | 6.81 | 5.89 | 3.03 | 0.87 | 0.59 |
| weight | 13.08 | 15.68 | 2.89 | 0.78 | 1.49 |

**Table 5:** Time ($s$)

| Dataset | OK $D = \mathrm{nnz}(\widehat{q})$ | Random $D = \mathrm{nnz}(\widehat{q})$ | Random $D = 10\,\mathrm{nnz}(\widehat{q})$ | ABMKSVM(ratio) | MKSVM |
|---|---|---|---|---|---|
| adult | 3.6 | 4.6 | 86.9 | 87.3 | 740.9 |
| reuters | 0.8 | 0.2 | 1.0 | 31207.4 | 17490.7 |
| buzz | 2.0 | 1.9 | 60.2 | 92.7 | 1035.1 |
| ads | 0.017 | 0.013 | 0.014 | 56.7 | 92.3 |
| farm | 0.27 | 0.05 | 8.3 | 86.3 | 180.0 |
| mnist17 | 3.4 | 4.0 | 53.1 | 38.0 | 702.6 |
| mnist49 | 3.7 | 4.4 | 78.1 | 27.0 | 602.5 |
| mnist56 | 2.9 | 3.6 | 56.4 | 24.3 | 623.9 |
| weight | 1.9 | 1.0 | 65.0 | 83.1 | 695.3 |

**Table 6:** Auxiliary experiments on mnist and weight with OK

| Dataset | Model | Base kernel | $\rho/N_w$ | %$n$ in problem (4) | Test error (%) | Time ($s$) |
|---|---|---|---|---|---|---|
| mnist17 | Ridge | Arc-cosine | 0.00100 | 50 | 1.06 | 9.1 |
| mnist49 | Ridge | Arc-cosine | 0.00100 | 50 | 1.91 | 9.4 |
| mnist56 | Ridge | Arc-cosine | 0.00100 | 50 | 1.68 | 8.3 |
| weight | Ridge | Gaussian | 0.00015 | 100 | 2.04 | 64.7 |

## Footnotes

[5] http://archive.ics.uci.edu/ml/datasets/Internet+Advertisements. We use all but the first 3 features which are sometimes missing in the data.

[6] https://archive.ics.uci.edu/ml/datasets/Farm+Ads

[7] http://yann.lecun.com/exdb/mnist/. We do pairwise classifications of digits 1 vs. 7, 4 vs. 9, and 5 vs. 6.

[8] http://archive.ics.uci.edu/ml/datasets/Weight+Lifting+Exercises+monitored+with+Inertial+Measurement+Units. We neglect the first 4 features, and furthermore we only use remaining features that are not missing in any datapoint. We consider classifying the datapoint as class A or not.