[Reviews · NeurIPS 2016]

Reviewer 1

Summary

The paper presents an approach for learning kernels from random features. The authors propose a two-stage approach, where first set of random features are drawn and then their weights are optimized to maximize the alignment to the target kernel. The authors show how to optimize the resulting model efficiently and provide theorems for consistency and generalization performance. The method is evaluated on some benchmark problems

Qualitative Assessment

The proposed method seems to provide a useful addition for kernel learning literature. In particular the capability to learn a better kernel when the original data is not well-aligned with the target (section 4.1) is potentially very useful. I am perhaps less enthusiastic about the feature selection experiments, since the compared method is selecting features completely randomly - we have better feature selection methods that that. The speedup over the joint optimization method is also notable. The write-up is slightly more convoluted than one would like and requires several reads to understand. Details: - line 64: you should state what assumptions you make about the feature function \phi - line 76 onwards: the notation could be described better, e.g. showing the dimensionalities of the objects e.g. W here - eq (6): I could not figure out why is the square root appearing here - line 169: It is not clear why all this follows from choosing the Gaussian kernel e.g. the exact form of the feature function probably is chosen independently?

Confidence in this Review

1-Less confident (might not have understood significant parts)


Reviewer 2

Summary

The authors present a supervised kernel learning method by random features. Starting from a kernel alignment problem, an efficient technique is developed and supported by theoretical analyses. They report on empirical results.

Qualitative Assessment

The technical contribution is strong. Also the writeup is clear. The empirical evaluation however leaves room for improvement although it is already comprehensive in terms of the studied settings. Nevertheless, it would be interested to compare the proposed approach with, say, standard MKL using a linear combination of many Gaussians (differently parameterized) or some other kernel learning technique. Another baseline that I am missing would be a Gaussian kernel where the bandwidth has been optimized by model selection techniques (cross-validation). In my view this would strengthen the argument in favor of the proposed method.

Confidence in this Review

1-Less confident (might not have understood significant parts)


Reviewer 3

Summary

This work presents a method to speed up kernel learning using random features (RF). With respect to known approaches, e.g. structured composition of kernels with respect to an alignment metric and joint kernel composition optimization based on empirical risk minimization, it achieves higher efficiency by optimizing compositions of kernels using explicit RF maps rather than full kernel matrices. The proposed method is divided in 2 steps: First, the kernel is learned efficiently using RF. Then, the optimized features associated to the learned kernel are used in a standard supervised learning setting to compute an estimator. Authors prove the consistency of the learned kernel and generalization guarantees for the learned estimator. An empirical evaluation of the method is also provided, including: - A toy example showing the learned features in the case of a bad initial kernel guess - An experiment involving a high-dimensional dataset, showing that the method attains higher predictive performance and induces a sparse representation, which can be potentially useful for interpretability - A performance evaluation on benchmark datasets, showing comparable test accuracies at a fraction of the computational cost with respect to the standard RF approach

Qualitative Assessment

The proposed approach is very interesting and novel. The idea of using random features to speed up kernel alignment is brilliant. The paper is well written and organized in a principled way. The authors provide a theoretical analysis of the method, guaranteeing consistency of the learned kernel and generalization properties of the resulting estimator. The experimental part allows to better visualize the method, shows how it can be used to identify sparse features in high dimensions and compares the accuracy and computational time of the proposed method on 3 benchmark datasets. Regarding the latter point, benchmarking experiments highlight promising practical applications for the method. As an improvement, I would suggest to evaluate the method on more benchmark datasets, for instance some of the ones used in [18]. Minor issues: 37 (?): optimization --> optimization problem 76, 83, 89: Properly introduce (or change) $W$ and $W^k$, highlighting the relationship with lower-case $w^i$ 87: need only store --> only needs to store 98: missing ")" the last element of $z^i$ 189: one-hot --> one-shot Algorithm 1: Add "end" at the end of each while cycle Figure 1: It is not clear why the optimized random features $w^k$ clustered around (-1,-1) and (1,1), depicted in yellow in the figure, would indicate a good solution.

Confidence in this Review

2-Confident (read it all; understood it all reasonably well)


Reviewer 4

Summary

Rahimi and Recht[21] formulation for binary classification problem is extended to (6). In (6), \hat{q} is inserted compared to (5). The \hat{q} is a solution of (4). (4) is also binary classification problem, but not include the regularizer.

Qualitative Assessment

There are not so much difference between Rahimi and Recht[21] But the generalisation error is better than[21]. Also computational speed is not but. The proposed modification is useful.

Confidence in this Review

1-Less confident (might not have understood significant parts)


Reviewer 5

Summary

The paper extended the random features for kernel machines work of Rahimi and Recht by developing an efficient optimization method allowing to learn a kernel. The empirical evaluations suggested its effectiveness.

Qualitative Assessment

It's not quite straightforward to follow the exact steps of the math, but the key parts seem make sense. The heuristic part of the method obviously influence the efficiency, but it hasn't been evaluated. The performance comparison only involves the previous method with three different kernels. I'd also like to see what would be the results if it includes other known efficient methods, since the paper claims the its general superiority.

Confidence in this Review

1-Less confident (might not have understood significant parts)